# Effects of mechanical weeding on soil fertility and microbial community structure in star anise (*Illicium verum* Hook.f.) plantations

Jian Xiao[1,2], Siyu Chen[2], Yan Sun[2], Siyu Wu[1], Wenhui Liang[1]\*, Shangdong Yang[2]\*

**1** Guangxi Forestry Research Institute/Guangxi Key Laboratory of Special Non-wood Forest Cultivation & Utilization/Engineering and Technology Research Center for Anise and Cinnamon of State Forestry and Grassland Administration/Guangxi Engineering and Technology Research Center for Woody Spices, Nanning, China, **2** Agricultural College, Guangxi University, Nanning, China

\* 723746615@qq.com (WL); 924433816@qq.com (SY)

**Data Availability Statement:** The raw reads were deposited into the NCBI Sequence Read Archive (SRA) database (Accession Number: SRP316160).

## Abstract

Recently, the effects of weed control on crop yield, quality and soil fertility have been increasingly investigated. However, soil microorganism diversity under weed control, especially for aromatic plants, is little studied. Mechanical weeding effects on soil fertility and microbial diversity in star anise plantations remain unknown, limiting improvements in crop quality and yield through weed control. Therefore, mechanical weeding (MW) and no weeding (NW) zones were randomly designed in the same star anise plantation to study the mechanical weeding impacts on soil biological properties and microbial diversity. The phosphatase activity of MW soil was significantly higher than that of NW soil; however, aminopeptidase activity was significantly lower than that under NW. There was no significant difference in $\beta$-glucosidase activity between MW and NW. Moreover, soil microbial biomass C and N in MW soil were significantly higher than those of NW, but soil microbial biomass P was significantly lower than that of NW. Proteobacteria, Acidobacteria, Actinobacteria, Chloroflexi, Planctomycetes, WPS-2, Firmicutes and Verrucomicrobia were the predominant bacterial phyla in MW and NW soils. Specifically, Bacteroidetes was enriched in MW soil, being the unique dominant bacteria. Ascomycota, Basidiomycota, unclassified_k_-Fungi, Rozellomycota and Mortierellomycota were the predominant fungi in MW and NW soils. The numbers of dominant bacterial genera (> 1%) were 26 and 23 for NW and MW soils, respectively. Among them, *norank_f__norank_o__norank_c__Subgroup*_6, 1921–2 and *norank_f__norank_o__B*12-*WMSP*1 went undetected in MW soil. Moreover, the numbers of dominant fungi in soils of star anise plantations were 11 and 9 for NW and MW, respectively. Among them, only *unclassified_f__Clavicipitaceae* and *Mortierella* went undetected in MW soils. Thus, soil microbial community structures are not significantly altered by mechanical weeding. The above results suggest that soil fertility can be improved and soil heath can be maintained by mechanical weeding in star anise plantations. Moreover, soil-borne diseases maybe easily occurred under NW treatment in star anise plantation.

**Funding:** This work was supported by Open Research Fund of Guangxi Key Laboratory of Special Non-wood Forest Cultivation & Utilization (19-B-02-01), Project of key R & D of Guangxi (No. AB1850014), and the eighth batch of special funds for specially invited experts in Guangxi Province, China. The funders had no role in study design, data collection and analysis, decision to publish, or preparation of the manuscript.

**Competing interests:** The authors have declared that no competing interests exist.

## Introduction

In agricultural systems, weeds are defined as plants growing in unwanted places and are related to crop damage through direct nutrition competition or parasitism. Moreover, weeds also cause indirect damage by harboring pests and crop pathogens. The direct losses caused by weeds are related to the different crop types and agroecological zones [1]. In China, more than 1,400 kinds of weeds have been reported, 30 of which are the most troublesome. More than 3 million tons of grain are lost every year due to weeds in China [2].

Star anise (*Illicium verum* Hook.f.) is an aromatic evergreen tree native to western China, Vietnam, Cambodia, Myanmar, Indonesia, the Philippines and other subtropical regions. Its fruits are not only an important Chinese medicine, treating vomiting, stomachache, insomnia, skin inflammation and rheumatism [3, 4], but also a common spice in cooking [5], being introduced into Europe in the 17th century. Its unique licorice flavors are the result of a compound called anisole [4, 6]. Moreover, its bark, seeds, leaves, branches and fruits are rich in fennel oil with antibacterial [7], antiviral [8], and antioxidant [9] functions. In the modern pharmaceutical industry, star anise is used as an industrial source of shikimic acid and an important ingredient of the antiviral drug Tamiflu (oseltamivir phosphate) [10].

Guangxi Province, located in southwestern China, is the native production area of star anise. The star anise yield in Guangxi accounts for 80% and more than 85% of its total production every year worldwide and throughout China, respectively [11]. However, the yield per unit area of star anise in Guangxi is very low and unstable due to inappropriate management, including weed control. Recently, this has become a limiting factor for the sustainable development of the star anise industry in Guangxi. As weeds interfere with tree growth at any time and any stage of production, they can compete for nutrients, water, space, light and even harbor pests and pathogens, causing problems for mechanized harvesting and potentially changing the final crop quality [12]. Therefore, weed control is an important problem to consider for the sustainable development of star anise production. To improve the yield and profit margins of the star anise industry, weeding in star anise plantations must be correctly established and properly maintained. That is, a good weed management practice is essential. Previous studies have shown that weeding is an effective practice for tree plantation management [13] because it promotes the growth of planted trees [14] by reducing competition between planted trees and weeds [15].

First, star anise growers mostly rely on the application of chemical herbicides and mechanical mowing in their production fields. However, frequent application of chemical herbicides not only causes negative environmental issues, such as herbicide leaching, drift and run-off, but also induces severe phytotoxic injuries to star anise trees, including stunted growth, chlorosis, burning and dropping of leaves, and even complete death of the tree. Due to these issues surrounding chemical herbicide use and increasing concern among star anise consumers related to herbicide and pesticide applications, many star anise producers are interested in strategies to reduce or even eliminate chemical weed management. Currently, mechanical mowing with a brush cutter is commonly conducted once or twice per year by operators in Guangxi star anise plantations. However, the effect of current weed control practices on soil quality, such as soil health and fertility in star anise plantations, has been seldom reported.

Soil quality is influenced by the interplay of physical, chemical and biological soil factors [16]. Soil physical and chemical indicators are commonly used by farmers and researchers to assess soil quality, and biological indicators are frequently underrepresented [17]. However, previous studies have shown that biological indicators, such as soil organisms, particularly soil microbial activity and community structure, can be sensitive to the quality and health of soil ecosystems [18]. In other words, biological indicators can help to better elucidate the soil

response to management as well as the underlying relationships. The purpose of this manuscript is to provide a theoretical direction for nonchemical weed control strategies in star anise plantations and identify areas where current practices could potentially support the sustainable development of star anise production.

## Materials and methods

### Study area and sampling

Soil samples were collected from the Paiyangshan forest farm (107˚5′E, 22˚1′N), Ningming County, Guangxi Zhuang Autonomous Region, Southwest China. This region is in the subtropical monsoon climate zone with an annual average temperature of 22 ˚C, and the mean annual precipitation rainfall ranges from 1400 to 2400 millimeters.

The soil type of the study area was acid red loam with a pH of 5.46 and an organic matter content of 12.9 g kg$^{-1}$. The total contents of nitrogen, phosphorus and potassium in the topsoil were 0.81, 0.39 and 2.68 g kg$^{-1}$, respectively. Moreover, the contents of alkaline nitrogen, available phosphorus and potassium were 53.7, 9.1 and 89.0 mg kg$^{-1}$, respectively.

Weed control approaches were arranged randomly with mechanical weeding (MW) and no weeding (NW) zones in the same star anise plantation. The experiment comprised three replicate blocks each with three plots (50 × 50 m each) assigned to two treatment combinations: MW and NW. Weeding with a brush cutter was conducted in early July 2019, and all the cut plant materials were left on the ground.

One month later, in August, soil samples were collected from the MW and NW blocks. First, we removed the litter on the soil surface and collected three disturbed soil subsamples from the 0–20 cm soil layer. Three samples were selected from each species in soil of star anise with different treatments, and one soil sample was taken from each tree. All soil samples were collected in separate sterile bags, which were sealed and placed in an incubator with ice packs until they were taken to the laboratory for processing. In the laboratory, the soil attached to the plant's fine roots was defined as rhizosphere soil, and the remaining soil was classified as bulk soil. The rhizosphere soil samples were used for DNA extraction and enzyme activity analysis, and the remaining soil samples were used for soil property determination.

### Soil biological property analysis

The activities of three soil enzymes involved in the C cycle ($\beta$-glucosidase), N cycle (aminopeptidase) and P cycle (phosphatase) were determined. The $\beta$-glucosidase activity was measured using the method described by Hayano [19]. Aminopeptidase activity was measured using the method described by Ladd [20]. Phosphatase activity was measured using the method described by Tabatabai and Bremner [21].

The soil microbial biomass carbon was determined by the chloroform fumigation extraction-volumetric analysis method [22], soil microbial biomass nitrogen was determined by ninhydrin colorimetry [23], and soil microbial biomass phosphorus was determined by the phosphorus molybdenum blue colorimetric method [24].

### Analysis of soil microbial diversity

Microbial community genomic DNA was extracted from samples using the FastDNA® Spin Kit for Soil (MP Biomedicals, U.S.) according to manufacturer′s instructions. The DNA extract was checked on a 1% agarose gel, and DNA concentration and purity were determined with a NanoDrop 2000 UV-vis spectrophotometer (Thermo Scientific, Wilmington, USA). The V3-V4 hypervariable region of the bacterial 16S rRNA gene was amplified with primer pairs

338F (5'-ACTCCTACGGGAGGCAGCAG-3') and 806R (5'-GGACTACHVGGGTWTCTAAT-3'), and the ITS1 region of the fungal ITS gene was amplified with primer pairs ITS1F (5'-CTTGGTCATTTAGAGGAAGTAA-3') and ITS2R (5'-GCTGCGTTCTTCATCGATGC-3') by an ABI GeneAmp® 9700 PCR thermocycler (ABI, CA, USA). PCR amplification of the 16S rRNA gene was performed as follows: initial denaturation at 95 ˚C for 3 min, followed by 27 cycles of denaturing at 95 ˚C for 30 s, annealing at 55 ˚C for 30 s and extension at 72 ˚C for 45 s, a single extension at 72 ˚C for 10 min, and a final extension at 4 ˚C. The PCR mixtures contained 5 × TransStart FastPfu buffer 4 μL, 2.5 mM dNTPs 2 μL, forward primer (5 μM) 0.8 μL, reverse primer (5 μM) 0.8 μL, TransStart FastPfu DNA Polymerase 0.4 μL, template DNA 10 ng, and ddH2O up to 20 μL. PCRs were performed in triplicate. The PCR products were extracted from a 2% agarose gel and purified using the AxyPrep DNA Gel Extraction Kit (Axygen Biosciences, Union City, CA, USA) according to the manufacturer's instructions and quantified using a Quantus™ Fluorometer (Promega, USA). Purified amplicons were pooled in equimolar amounts and paired-end sequenced (2 × 300) on an Illumina MiSeq platform (Illumina, San Diego, USA) according to the standard protocols by Majorbio Bio-Pharm Technology Co. Ltd. (Shanghai, China). The raw reads were deposited into the NCBI Sequence Read Archive (SRA) database (Accession Number: SRP316160).

The Quantitative Insights Into Microbial Ecology (Qiime version 1.9.1) pipeline was used to process the sequences [25]. The raw 16S and ITS rRNA gene sequencing reads were demultiplexed, quality-filtered by Trimmomatic and merged by FLASH with the following criteria: (i) the 300 bp reads were truncated at any site receiving an average quality score of <20 over a 50 bp sliding window, and the truncated reads shorter than 50 bp were discarded, reads containing ambiguous characters were also discarded; (ii) only overlapping sequences longer than 10 bp were assembled according to their overlapped sequence. The maximum mismatch ratio of overlap region is 0.2. Reads that could not be assembled were discarded; (iii) Samples were distinguished according to the barcode and primers, and the sequence direction was adjusted, exact barcode matching, 2 nucleotide mismatch in primer matching. In summary, labelled barcodes, low-quality reads, and ambiguous nucleotides were removed to obtained high-quality sequences. Operational taxonomic units (OTUs) with 97% similarity cutoff were clustered using UPARSE (version 7.1, http://drive5.com/uparse/), and chimeric sequences were identified and removed [26]. The taxonomy of each OTU representative sequence was analyzed by RDP Classifier (http://rdp.cme.msu.edu/) against the SILVA (v132) SSU and the UNITE (v7.2) reference databases using confidence threshold of 0.7.

## Statistical analyses

Alpha diversities of bacterial and fungal communities were calculated using Mothur (version v.1.30.2, https://mothur.org/wiki/calculators/). Principal Component Analysis (PCA) was performed, and the R language (version 3.3.1) tool was used for statistics and graphing. For microbial community composition and Venn diagram analysis, OTU tables with a 97% similarity level were selected, and the R language (version 3.3.1) tool was used for statistics and graphing. Linear discriminant analysis (LDA) was performed using LEfSe (http://huttenhower.sph.harvard.edu/galaxy/root?tool_id=lefse_upload) on samples according to different grouping conditions based on taxonomic composition to identify clusters that had a significant differential impact on sample delineation. Functional predictions of the soil fungal communities by Fungi Functional Guild (FUN Guild) tool. The Wilcoxon rank-sum test was used to analyze the significant differences in fungal functions.

All treatments were performed in triplicate with a completely random design. The mean values of soil enzyme activities, microbial biomass, soil bacterial and fungal diversities and

richness were compared by Student's t-test using Statistical Product and Service Solutions (SPSS version 26) software with a significance level of 0.05. The results are shown as the standard deviation of the mean (mean ± SD). The experimental data were analyzed by using Excel 2019 and SPSS Statistics 21, and online data analysis was conducted by using the free online Majorbio Cloud Platform (www.majorbio.com) of Majorbio Bio-Pharm Technology Co. Ltd. (Shanghai, China).

## Results

### Soil enzyme activities

The changes in soil enzyme activities between the MW and NW treatments in the star anise plantation are shown in Table 1. The activities of phosphatase and aminopeptidase in the MW treatment were significantly different from those in the NW treatment. However, even though the activity of aminopeptidase in the NW soil was significantly higher than that in the MW soil, the activity of phosphatase in the soil of the NW treatment showed the opposite trend as aminopeptidase, i.e., the activity of phosphatase in the soil of the MW treatment was significantly higher than that of the NW soil. This result suggested that the phosphorus cycling process in soil can be promoted by mechanical weeding in star anise plantations (Table 1).

### Soil microbial biomass

As seen in Table 2, the soil microbial biomass carbon (SMC) and nitrogen (SMN) in the MW treatment were all significantly higher than those in the NW treatment, and only the soil microbial biomass phosphorus (SMP) in the MW treatment was significantly lower than that in the NW treatment. This result indicates that soil microbial biomass carbon, nitrogen and phosphorus can all be significantly changed by mechanical weeding in star anise plantations. That is, soil fertility in star anise plantations can be significantly changed by mechanical weeding (Table 2).

### Soil bacterial and fungal diversity and richness

As seen in Table 3, the coverage indexes were all 99% and above, it indicates that the analytical data in Table 3 are reliable. The Shannon index, which was used as indicators of bacterial and

**Table 1. Soil enzyme activities between NW and MW treatments in star anise plantations (n mol/g min$^{-1}$ 30°C).**

| Samples | $\beta$-Glucosidase | Aminopeptidase | Phosphatase |
|---|---|---|---|
| No weeding (NW) | 0.27±0.02a | 16.05±0.50a | 0.13±0.02b |
| Mechanical Weeding (MW) | 0.29±0.02a | 12.98±0.30b | 0.34±0.05a |

*Note*. All data are presented as the means ± SD (standard deviation). The Student's t-test was performed ($P < 0.05$). Different letters in the same column indicate significant differences among treatments at $P < 0.05$.

**Table 2. Soil microbial biomass between the NW and MW treatments in star anise plantations (mg kg$^{-1}$).**

| Samples | Microbial biomass C | Microbial biomass N | Microbial biomass P |
|---|---|---|---|
| No weeding (NW) | 13.10±1.87b | 3.54±0.19b | 29.47±1.47a |
| Mechanical Weeding (MW) | 194.01±6.09a | 22.52±1.98a | 14.92±1.49b |

*Note*. All data are presented as the means ± SD (standard deviation). The Student's t-test was performed ($P < 0.05$). Different letters in the same column indicate significant differences among treatments at $P < 0.05$.

**Table 3. Alpha diversity index of soil bacteria and fungi in star anise plantations between the NW and MW treatments (OTU level).**

|  | Treatments | Shannon index | Ace index | Chao1 index | Coverage |
|---|---|---|---|---|---|
| Bacteria | No weeding (NW) | 6.03±0.33a | 2414.98±244.60a | 2372.22±263.27a | 0.99 |
|  | Mechanical Weeding (MW) | 5.90±0.11a | 2468.82±85.85a | 2443.81±116.81a | 0.99 |
| Fungi | No weeding (NW) | 4.03±0.32a | 806.88±42.97a | 811.17±42.26a | 1.00 |
|  | Mechanical Weeding (MW) | 3.36±0.81a | 793.08±133.08a | 789.91±132.88a | 1.00 |

*Note*. All data are presented as the means ± SD (standard deviation). The Student's t-test was performed ($P < 0.05$). Different letters in the same column indicate significant differences among treatments at $P < 0.05$.

fungal diversities, no significant differences could be found between NW and MW treatments. In addition, the Ace and Chao1 indexes, which were used as indicators of microbial richness, were also no significant difference between NW and MW treatments. To evaluate the extent of the similarity of the soil bacterial and fungal communities. unweighted UniFrace Principal Component Analysis (PCA) at OUT level was performed. The results suggested that there were quite similarity with the bacterial and fungal compositions between NW and MW (Fig 1). All above results suggested that soil microbial diversity and richness in star anise plantations were not significant changes by mechanical weeding.

## Analysis of soil microbial community structure in star anise plantations between the NW and MW treatments

As seen in Fig 2A, 8 and 9 dominant bacteria with relative abundances greater than 1% could be detected in soils of NW and MW at the phylum level, respectively. In soils of star anise plantations under the NW treatment, Proteobacteria (34.72%), Acidobacteria (18.63%), Actinobacteria (16.52%), Chloroflexi (16.85%), Planctomycetes (2.57%), WPS-2 (2.08%), Firmicutes (2.19%), Verrucomicrobia (1.75%) and others (3.72%) were the dominant bacteria at the phylum level. In contrast, Proteobacteria (36.06%), Acidobacteria (17.69%), Actinobacteria (19.72%), Chloroflexi (15.42%), Planctomycetes (2.42%), WPS-2 (2.02%), Firmicutes (1.18%), Verrucomicrobia (1.48%), Bacteroidetes (1.17%) and others (2.85%) were the dominant

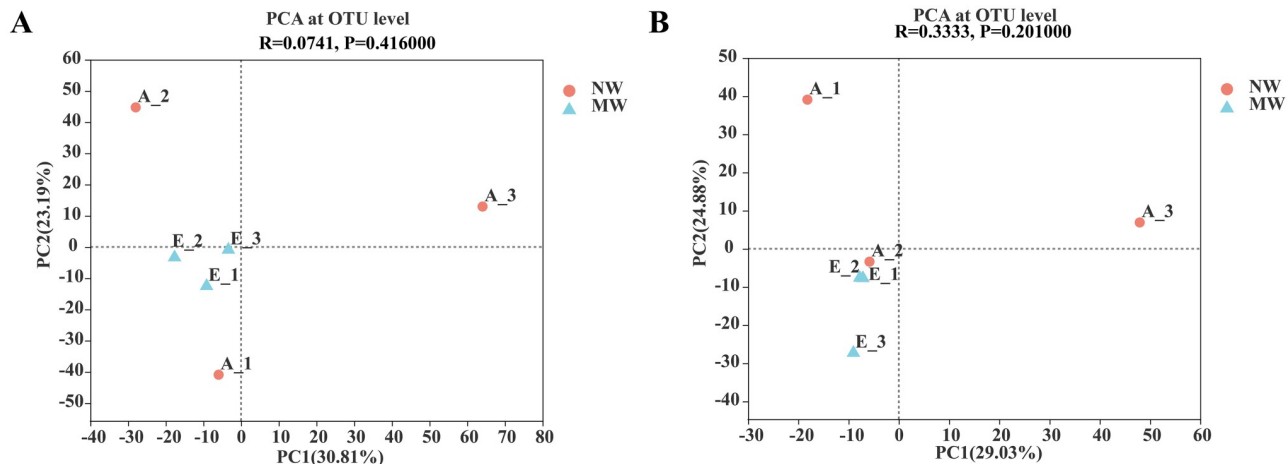

**Fig 1. PCA of soil bacterial (A) and fungal (B) communities in star anise plantations between the NW and MW treatments (OTU level).** *Note.* NW: no weeding in the star anise plantation, and MW: mechanical weeding in the star anise plantation. ANOSIM analysis was used to test differences between groups.

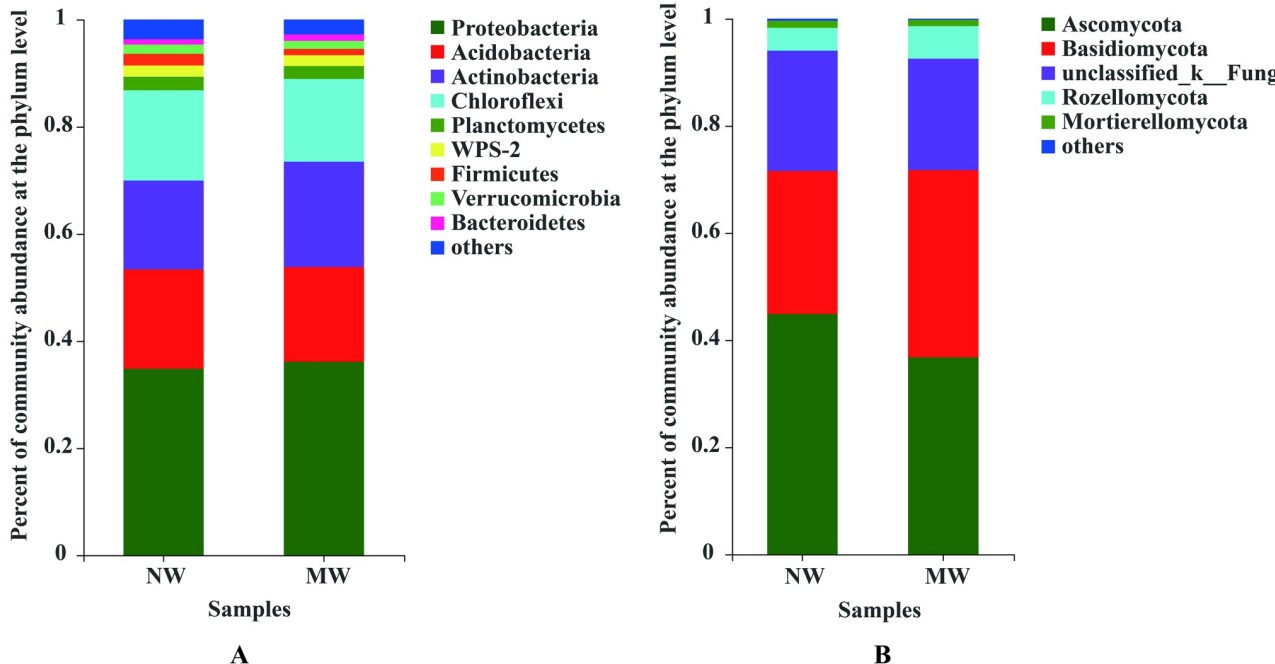

**Fig 2. Compositions of soil bacterial (A) and fungal (B) communities at the phylum level between the NW and MW treatments.** *Note.* Only the phyla with relative abundances greater than 1% are presented in Fig 2. NW: no weeding in the star anise plantation, and MW: mechanical weeding in the star anise plantation.

bacteria at the phylum level in the soil under the MW treatment. The results showed that not only were the proportions of dominant bacteria at the phylum level changed, i.e., the proportions of Acidobacteria, Chloroflexi, Planctomycetes, WPS-2, Firmicutes, and Verrucomicrobia decreased, and those of Proteobacteria and Actinobacteria increased, but Bacteroidetes were also enriched as dominant bacteria in star anise plantations under MW treatment Fig 2A.

Moreover, all 5 dominant fungal phyla with relative abundances greater than 1% could be found in the soil of star anise plantations under the NW and MW treatments. In soils under the NW treatment, Ascomycota (44.90%), Basidiomycota (26.71%), unclassified_k_Fungi (22.44%), Rozellomycota (4.24%) and Mortierellomycota (1.33%) could be detected. In contrast, Ascomycota (36.73%), Basidiomycota (34.96%), unclassified_k_Fungi (20.80%), Rozellomycota (6.12%) and Mortierellomycota (1.17%) could be found in the soil under the MW treatment. These results suggested that the soil fungal community composition in star anise plantations is not altered by mechanical weeding. However, the proportion of Ascomycota, unclassified_k_fungi and Mortierellomycota decreased, and Basidiomycota and Rozellomycota increased in the MW treatment compared to that in the NW treatment Fig 2B.

As shown in Fig 3, the numbers of dominant bacteria (> 1%) in the star anise plantation soil were 26 and 23 for the NW and MW treatments, respectively. *norank_f__norank_o__norank_c__Subgroup_6*, *1921–2* and *norank_f__norank_o__B12-WMSP1* were the three genera that were undetected in the soil of the MW treatment. Moreover, the proportions of mutually dominant bacteria in soils of the NW and MW treatments, such as *norank_f__norank_o__Subgroup_2*, *norank_f__norank_o__Acidobacteriales*, *norank_f__norank_o__norank_c__AD3*, *unclassified_f__Ktedonobacteraceae*, *norank_f__norank_o__norank_c__norank_p__WPS-2*, *Burkholderia-Caballeronia-Paraburkholderia*, *norank_f__Gemmataceae*, *Conexibacter*, and *Mycobacterium*, were all decreased in soil of the MW treatment compared to those under the

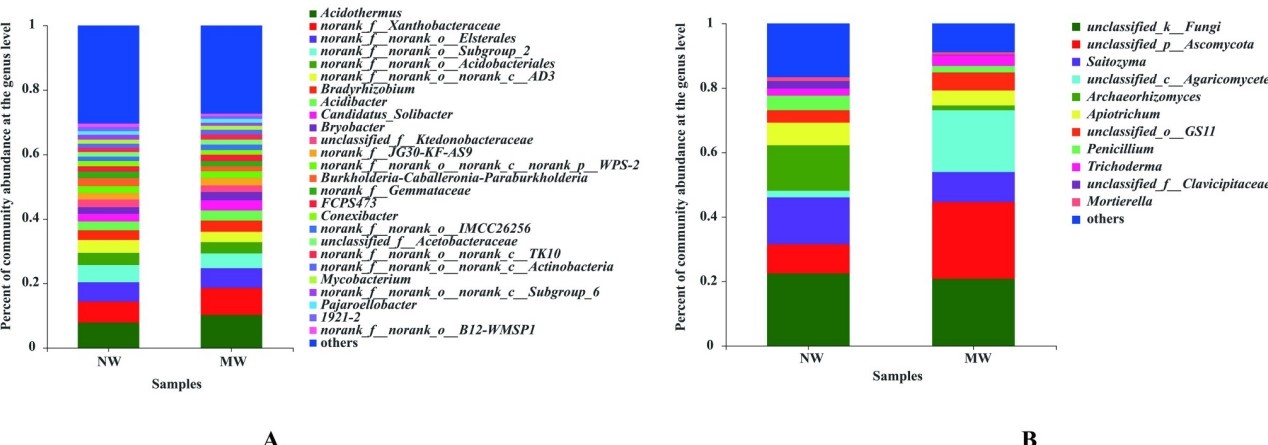

**Fig 3. Compositions of soil bacterial (A) and fungal (B) communities at the genus level in star anise plantations between the NW and MW treatments.** *Note.* Only the genera with relative abundances greater than 1% are presented in Fig 3. NW: no weeding in the star anise plantation, and MW: mechanical weeding in the star anise plantation.

NW treatment (S1 Table). This suggests that the bacterial community structure in the soil of star anise plantations can be altered by mechanical weeding management.

In addition, the numbers of dominant fungi in the soil of star anise plantations were 11 and 9 for the NW and MW treatments, respectively. There were 9 mutual fungal genera in the NW and MW treatments, while *unclassified_f__Clavicipitaceae* and *Mortierella* were not detected in the soil of the MW treatment. Moreover, the proportions of mutual fungal genera, such as *unclassified_k__Fungi*, *Saitozyma*, *Archaeorhizomyces*, *Apiotrichum* and *Penicillium*, were all decreased in the soil of the MW treatment compared to that under the NW treatment (S2 Table). The unique dominant fungi under NW were *unclassified_f__Clavicipitaceae* and *Mortierella*. The above results also suggest that the soil fungal community structure in star anise plantations was changed by the MW treatment.

As seen in Fig 4, there were 405 mutually dominant bacteria at the genus level in the soil of the star anise plantation between the NW and MW treatments, but there were 50 and 27 unique dominant bacteria in the soil of the star anise plantation between the NW and MW treatments, respectively Fig 4A. Moreover, there were 164 mutually dominant fungi at the genus level in the soil of star anise plantations between the NW and MW treatments, but there were 98 and 39 unique dominant fungi in the soil of star anise plantations between the NW and MW treatments, respectively Fig 4B.

These results also suggest that soil bacterial and fungal community structures are all altered by mechanical weeding. In particular, the amounts of dominant bacteria or fungi were all decreased by the mechanical treatment.

## Statistical differences of microbial communities at different levels

As analysis of the microbial communities at OTUs level in this study by LEfSe would be too complex. Therefore, microbial community structure at phylum and genus levels only were described in this paper.

As seen at Figs 5 and 6, at phylum level, two phyla of bacteria and one phylum of fungi in NW treatment were significantly enriched, namely Gemmatimonadetes, Elusimicrobia and Glomeromycota, respectively. By contrast, there were no any significantly changed of bacteria and fungi at phylum level in MW treatment. In addition, 5 bacterial genera, such as *norank_f__Gemmatimonadaceae*, *Inquilinus*, *Actinomadura*, *Silvimonas*, significantly enriched in

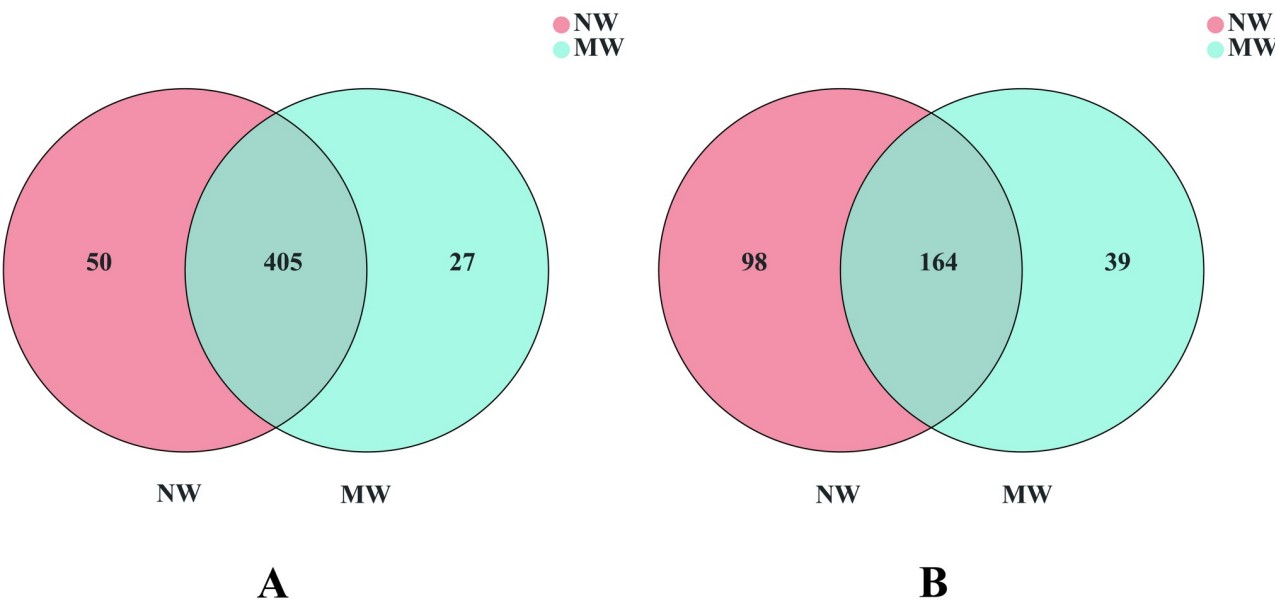

**Fig 4. Venn Diagram of soil bacterial (A) and fungal (B) communities at genus level between the NW and MW treatments in star anise plantations.** *Note*. NW: no weeding in the star anise plantation, and MW: mechanical weeding in the star anise plantation.

NW treatment, Meanwhile, 23 fungal genera, i.e., *norank_f__P3OB-42*, *Archaeorhizomyces*, *unclassified_f__Clavicipitaceae*, *unclassified_f__Chaetomiaceae*, *unclassified_f__Myxotricha-ceae*, *unclassified_f__Aspergillaceae*, *Talaromyces*, *unclassified_f__Chrysozymaceae*, *Cladophia-lophora*, *unclassified_o__GS33*, *Purpureocillium*, *Trechispora*, *Xylogone*, *Neopestalotiopsis*, *Pestalotiopsis*, *Speiropsis*, *unclassified_f__Agaricaceae*, *Pseudeurotium*, *unclassified_f__Chae-tothyriaceae*, *Scedosporium*, *unclassified_f__Sarcosomataceae*, *unclassified_o__Sordariales*, *Curvularia*, *Verticillium* were also significantly increased in NW treatment, too. On the con-trary, 8 bacterial genera, such as *Aquisphaera*, *norank_f__norank_o__WD260*, *Microbacter-ium*, *unclassified_o__Babeliales*, *Uliginosibacterium*, *Microbispora*, *Rudaea*, *Microvirga* and 2 fungal genera, namely *unclassified_f__Cuniculitremaceae* and *Inaequalispora* significantly increased in MW treatment only.

## Functional predictions of the soil fungal communities by FUNGuild

Fungi Functional Guild (FUN Guild) was employed to predict fungal functions in the two treatments. The Wilcoxon rank-sum test was used to analyze the significant differences in fun-gal functions between NW and MW. Among them, only the guild which related to unknown of soil fungi in MW treatment could be found higher than those of NW treatment, but there was no significant difference (Fig 7). Moreover, the guilds related to Soil Saprotroph, Fungal Parasite-Undefined Saprotroph, Undefined Saprotroph, Endophyte-Litter Saprotroph-Soil Saprotroph-Undefined Saprotroph, Animal Pathogen-Dung Saprotroph-Endophyte-Epi-phyte-Plant Saprotroph-Wood Saprotroph of soil fungi in MW treatment were lower than those of NW treatment, but there were no significant differences.

## Discussion

The enzymatic activity in soil is mainly derived from microbial origins, such as intracellular, cell-associated or free enzymes, which play an important role in maintaining soil health and the environment [27]. Soil enzymes can play key biochemical functions through the

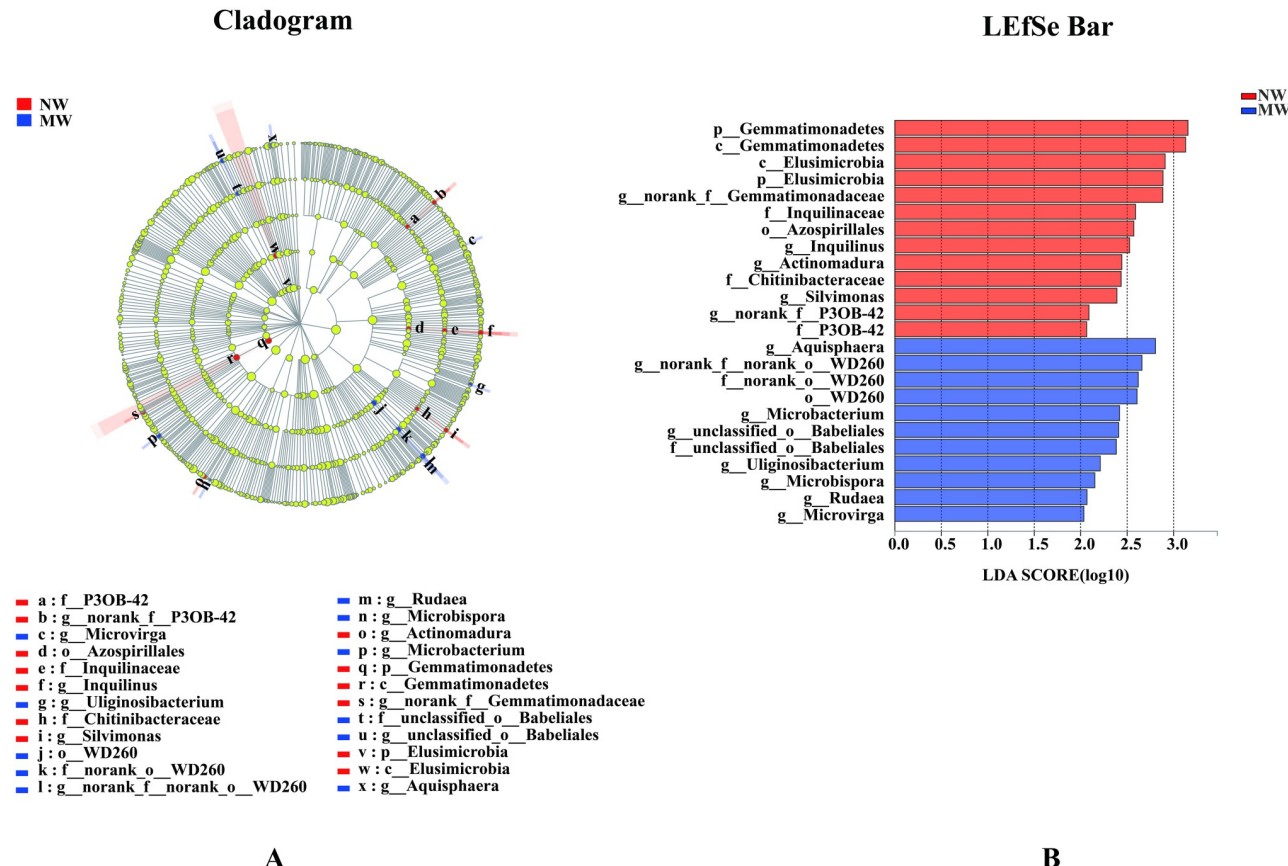

**Fig 5.** Cladogram showing the phylogenetic distribution of the bacterial lineages associated with soil from two treatments (A). Indicator bacteria with LDA scores of 2 or greater in bacterial communities associated with soil from the two treatments (B). Different colour regions represent different constituents (Red: NW; Blue, MW). Circles indicate phylogenetic level from phylum to genus. The diameter of each circle is proportional to the abundance of the group. Different prefixes indicate different levels (p: phylum; c: class; o: Order; f: Family; g: Genus). *Note*. NW: no weeding in the star anise plantation, and MW: mechanical weeding in the star anise plantation.

decomposition of organic matter, catalyzing microbial life processes, and stabilizing soil structure and nutrient cycling in the soil system [27–29].

In this experiment, we found that the activity of *β*-glucosidase was not significantly different between the MW and NW treatments, but the activity of phosphatase in soils of the MW treatment was significantly higher than that of the NW treatment. The activity of aminopeptidase in the soil of the MW treatment showed the opposite trend as the activity of phosphatase. This result suggested that soil fertility and health were not significantly altered by the MW treatment compared to the NW treatment.

Soil microbial biomass is also an important indicator of soil quality and functions in maintaining soil fertility and crop productivity [24]. The greater the microbial biomass in the soil, the greater the capacity of the soil to provide nutrients to plants through mineralization of organic nutrients [30]. Among them, soil microbial biomass carbon can not only promote the formation of new humus with high activity in soil but can also reflect the slight changes in soil before changes in the soil total carbon content [31]. Soil microbial biomass nitrogen can reflect the availability of soil nitrogen and play an important role in the supply and circulation of soil nitrogen [31]. Soil microbial biomass phosphorus can reflect the supply level of soil phosphorus [32]. In addition, although microbial biomass phosphorus cannot be directly absorbed and

**Cladogram**

**LEfSe Bar**

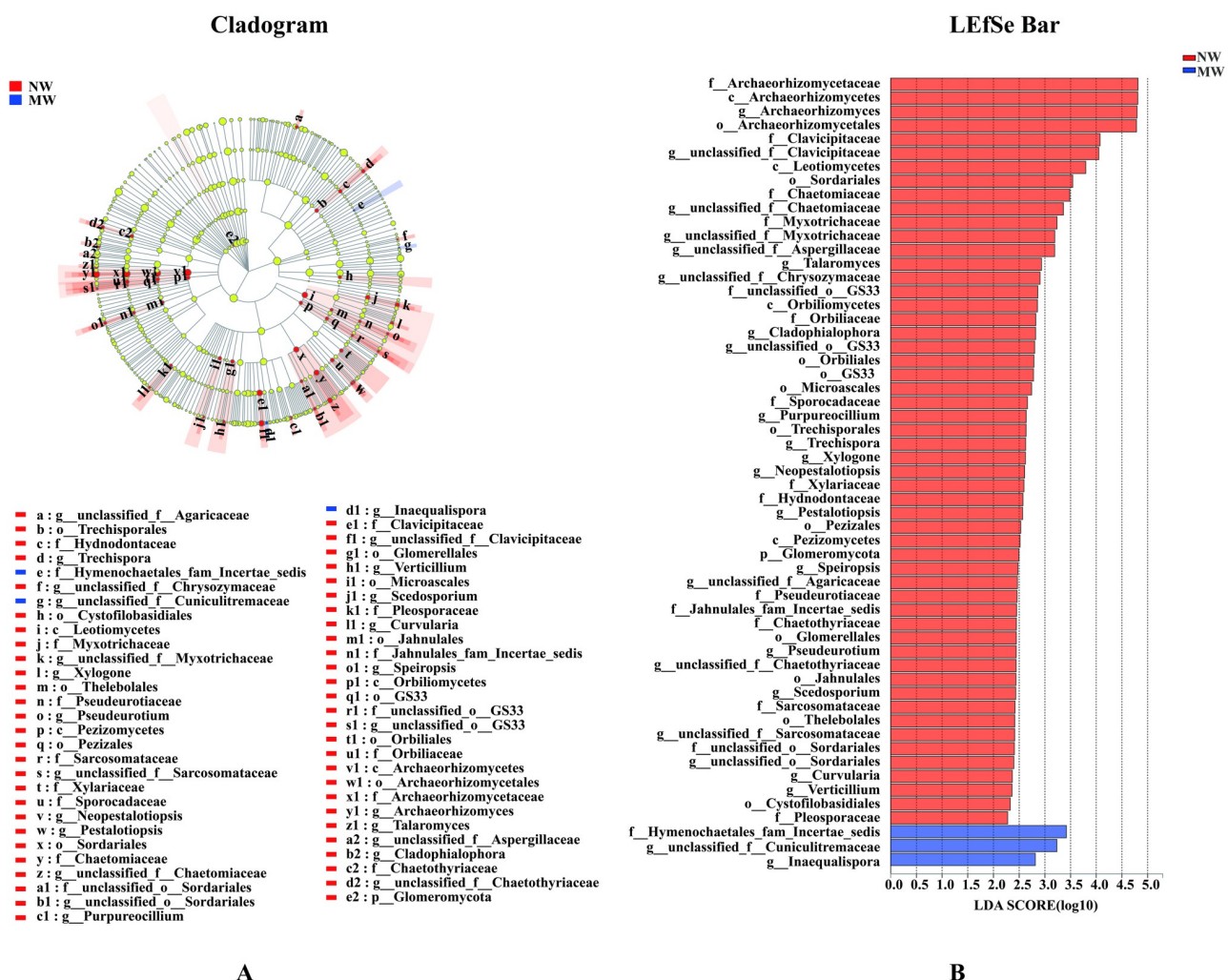

**Fig 6.** Cladogram showing the phylogenetic distribution of the fungal lineages associated with soil from two treatments (A). Indicator fungi with LDA scores of 2 or greater in fungal communities associated with soil from the two treatments (B). Different colour regions represent different constituents (Red, NW; Blue, MW). Circles indicate phylogenetic level from phylum to genus. The diameter of each circle is proportional to the abundance of the group. Different prefixes indicate different levels (p: phylum; c: class; o: Order; f: Family; g: Genus). *Note*. NW: no weeding in the star anise plantation, and MW: mechanical weeding in the star anise plantation.

utilized by plants, the turnover of microbial biomass phosphorus can slowly release inorganic phosphorus, so it has always been considered a source of available phosphorus in the soil, which is very important for plant growth [33]. Some studies have shown that weeding is more effective than nitrogen application for cilantro, especially under poor soil conditions [34]. We also found that biomass C and N in the soil of the MW treatment were significantly higher than those under the NW treatment. However, the microbial biomass P in the MW soil was significantly lower than that in the NW soil. As some types of weeds have good phosphorus uptake capacity, which can accumulate soil surplus phosphorus in biomass and promote phosphorus biocycling [35]. However, without total topsoil inversion, the plant materials cut by a brush cutter and incorporation into the soil which provide fertility is limited. A stratified layer of phosphorus builds up on or near the soil surface, increasing the risk of phosphorus release and subsequent transport of dissolved reactive phosphorus via surface runoff and tile drain water [36]. Therefore, it can be inferred that weed management by mechanical weeding in star

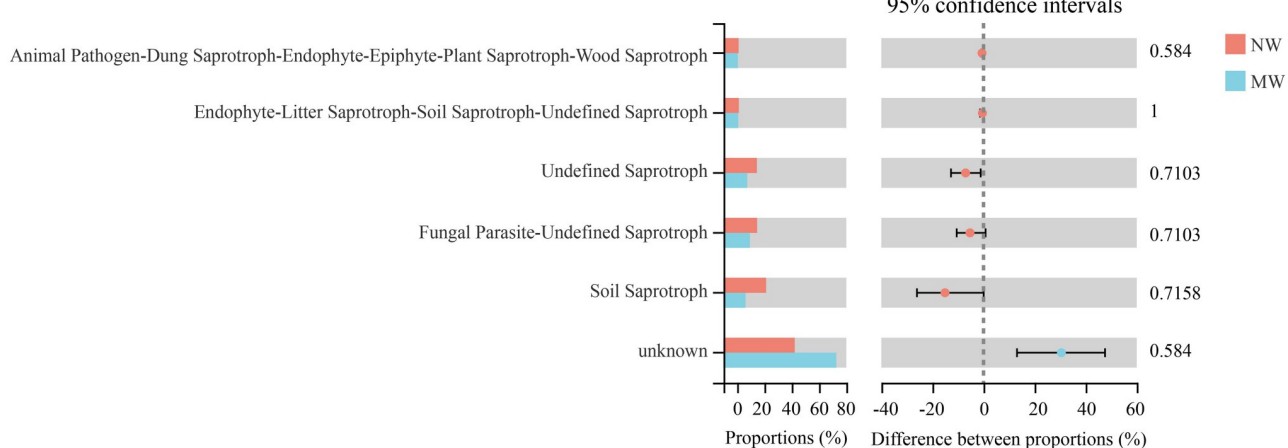

**Fig 7. Functional predictions of the soil fungal communities between the NW and MW treatments.** *Note*. NW: no weeding in the star anise plantation, and MW: mechanical weeding in the star anise plantation. The Wilcoxon rank-sum test was performed ($P < 0.05$).

anise plantations is not completely harmful to soil fertility and health. Nevertheless, the plant materials need to be smashed and mixed with the soil after weeding to prevent phosphorus loss. In contrast, mechanical weeding promoted soil organic carbon and nitrogen but not soil organic phosphorus in the star anise plantation.

Bacteria are the most abundant and diverse group of soil microorganisms [37]. They play a crucial role in terrestrial ecosystems, including mineralization of dead organic matter, binding of humic compounds in the soil mineral layer, carbon and nitrogen cycling, and provision of nutrients for plant growth [38–40]. Soil fungi are present in forest ecosystems as decomposers and are involved in material cycling, energy flow and information transfer [41, 42]. Among them, symbiotic fungi are important components of soil microbiomes that form mutually beneficial symbiotic relationships with plant roots, thus facilitating the uptake of nutrients such as nitrogen and phosphorus by plants [43], increasing their resistance to stress, pests and diseases [44, 45], contributing to the formation of soil aggregates and improving soil structure [46]. Soil bacteria and fungi are sensitive to changes in soil quality and are often used to assess environmental changes in ecosystems [47]. Furthermore, soil microbes also play a key role in ecosystem nutrient cycling, affecting vegetation development and succession [48].

In our study, at the phylum level, the proportions of Acidobacteria, Chloroflexi, Planctomycetes, WPS-2, Firmicutes and Verrucomicrobia decreased, but the proportions of Proteobacteria and Actinobacteria increased in soils in the MW treatment compared to those in the NW treatment. Previous studies have confirmed that Proteobacteria are copiotrophic microorganisms that thrive under conditions of high nutrient availability [49]. In addition, Actinomycetes play a key role in terrestrial ecosystem functioning by contributing to the global carbon cycle through the decomposition of soil organic matter, increasing plant productivity, and being producers of bioactive compounds essential for human, animal and environmental health [50]. Moreover, the composition and proportions of the soil fungal community at the phylum level were not significantly altered by the MW treatment compared to the NW treatment. This result suggested that soil microbial community structures at the phylum level were not significantly altered in soils of star anise plantations by mechanical weeding management.

At the genus level, the soil bacterial community structure could be altered by mechanical weeding management, with *Acidothermus*, *norank_f__Xanthobacteraceae*, *norank_f__norank_o__Elsterales*, *Bradyrhizobium*, *Acidibacter*, *Candidatus_Solibacter*, *Bryobacter*, *norank_f__JG30-KF-AS9*, *FCPS473*, *norank_f__norank_o__IMCC26256*, *unclassified_f__Acetobacteraceae*, *norank_f__norank_o__norank_c__TK10*, *norank_f__norank_o__norank_c__Actinobacteria*, *Pajaroellobacter*, *increasing*, *norank_f__norank_o__Subgroup_2*, *norank_f__norank_o__Acidobacteriales*, *norank_f__norank_o__norank_c__AD3*, *unclassified_f__Ktedonobacteraceae*, *norank_f__norank_o__norank_c__norank_p__WPS-2*, *Burkholderia-Caballeronia-Paraburkholderia*, *norank_f__Gemmataceae*, *Conexibacter*, *Mycobacterium* decreasing in soils of the MW treatment. Moreover, *norank_f__norank_o__norank_c__Subgroup_6*, 1921–2 and *norank_f__norank_o__B12-WMSP1* became the unique dominant bacteria in the soil of the NW treatment only. For the fungal community structure, even though the compositions were not significantly changed, the proportions were altered by mechanical weeding management. That is, *unclassified_p__Ascomycota*, *unclassified_c__Agaricomycetes*, *unclassified_o__GS11* and *Trichoderma* increased, and *unclassified_k__Fungi*, *Saitozyma*, *Archaeorhizomyces*, *Apiotrichum*, *Penicillium* decreased in soils under the MW treatment compared to those under the NW treatment. Moreover, *unclassified_f__Clavicipitaceaeand* and *Mortierella* became the unique dominant fungus at the genus level in soils of the NW treatment. As *Mortierella* was known as a functional microbe in assisting crops and mycorrhizal fungi in phosphorus (P) acquisition. And inorganic P from a range of soils could be dissolved effectively by several members of *Mortierella* by synthesizing and secreting oxalic acid [51]. Therefore, in association with the microbial biomass P, the ability of soil P supply under NW treatment could be considered higher than that of MW treatment in star anise plantation. However, some plant pathogens, which belongs to the family of Clavicipitaceae [52], enriched as unique dominant fungus in soil of NW treatment. It also suggested that soil-borne diseases maybe easily occurred under NW treatment in star anise plantation.

These results suggested that even though the soil microbial community structure, including bacterial and fungal compositions at the genus level, could be altered by mechanical weeding mainly via changes in the proportions of dominant bacteria and fungi, the compositions of bacteria and fungi in soils of the MW treatment were less changed than those in soils of the NW treatment. It can also be concluded that soil fertility and health in star anise plantations can be improved or maintained by mechanical weeding management.

## Conclusions

In this study, a field experiment was carried out to elucidate the effects of mechanical weeding on soil fertility and health in a star anise (*Illicium verum* Hook.f.) plantation. The conclusions were as follows. The activity of phosphatase in the soil of the MW treatment was significantly higher than that of the NW treatment. However, the activity of aminopeptidase was significantly lower than that of the NW treatment, and there was no significant difference in the activity of *β*-glucosidase between the MW and NW treatments. Soil microbial biomass C and N in the soil of the MW treatment were significantly higher than those of the NW treatment, but the soil microbial biomass P was significantly lower than that of the NW treatment. Proteobacteria, Acidobacteria, Actinobacteria, Chloroflexi, Planctomycetes, WPS-2, Firmicutes and Verrucomicrobia were the common dominant bacteria at the phylum level in soils of the MW and NW treatments. In particularly, Bacteroidetes was enriched in the soil of the MW treatment as the unique dominant bacteria. Ascomycota, Basidiomycota, unclassified_k_-Fungi, Rozellomycota and Mortierellomycota were the common dominant fungi at the phylum level in soils of the MW and NW treatments. The numbers of dominant soil bacteria at

the genus level in star anise plantations between the NW and MW treatments were 26 and 23, respectively. *norank_f__norank_o__norank_c__Subgroup_6*, *norank_f__norank_o__B12-WMSP1* and *1921–2* were the three bacterial genera that were undetected in the soils of the MW treatment. The numbers of dominant soil fungi at the genus level in the star anise plantation between the NW and MW treatments were 11 and 9, respectively. *Mortierella* and *unclassified_f__Clavicipitaceae* were undetected in soils of the MW treatment. The above results suggest that soil fertility can be improved and that soil heath in star anise plantations can be maintained by the mechanical weeding method. Moreover, soil-borne diseases maybe easily occurred under NW treatment in star anise plantation.

## Supporting information

**S1 Table. Proportions of soil dominant bacterial communities at genus level between the NW and MW treatments in star anise plantations (%).**
(DOCX)

**S2 Table. Proportions of soil dominant fungal communities at genus level between the NW and MW treatments in star anise plantations (%).**
(DOCX)

## Acknowledgments

The authors are also thankful to American Journal Experts (AJE) for editing services.

## Author Contributions

**Conceptualization:** Wenhui Liang, Shangdong Yang.

**Data curation:** Jian Xiao.

**Formal analysis:** Jian Xiao.

**Funding acquisition:** Wenhui Liang, Shangdong Yang.

**Methodology:** Siyu Chen, Yan Sun.

**Software:** Siyu Wu.

**Writing – original draft:** Jian Xiao.

**Writing – review & editing:** Jian Xiao, Shangdong Yang.

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
