## [Decision Letter · Decision Letter 0]

20 Dec 2021

PONE-D-21-29819Effects of mechanical weeding on soil fertility and microbial community structure in star anise (Illiciumverum Hook.f.) plantationsPLOS ONE

Dear Dr. Yang,

Thank you for submitting your manuscript to PLOS ONE. After careful consideration, we feel that it has merit but does not fully meet PLOS ONE’s publication criteria as it currently stands. Therefore, we invite you to submit a revised version of the manuscript that addresses the points raised during the review process.

Both reviewers felt the premise of this study is interesting. However, both noted that there appears to be an overall lack of rigor in the experimental setup and data analysis performed. In particular, there must be more detailed statistical analysis and additional analyses performed to support the conclusions made.

We look forward to receiving your revised manuscript.

Kind regards,

Brenda A Wilson, Ph.D.

Academic Editor

PLOS ONE

Journal Requirements:

“This work was supported by Open Research Fund of Guangxi Key Laboratory of Special Non-wood Forest Cultivation & Utilization (19-B-02-01), Project of key R & D of Guangxi (No. AB1850014), and the eighth batch of special funds for specially invited experts in Guangxi Province, China.”

Additional Editor Comments:

There appears to be an overall lack of rigor in the experimental setup and data analysis performed. In particular, the authors must include more detailed statistical analysis and additional analyses performed to support the conclusions made.

Reviewers' comments:

Reviewer's Responses to Questions

**Comments to the Author**

1. Is the manuscript technically sound, and do the data support the conclusions?

Reviewer #1: Partly

Reviewer #2: Partly

2. Has the statistical analysis been performed appropriately and rigorously? 

Reviewer #1: No

Reviewer #2: No

3. Have the authors made all data underlying the findings in their manuscript fully available?

Reviewer #1: Yes

Reviewer #2: No

4. Is the manuscript presented in an intelligible fashion and written in standard English?

Reviewer #1: Yes

Reviewer #2: Yes

5. Review Comments to the Author

Reviewer #1: The manuscript is well organized and written but there are some major concerns that need to be addressed in the manuscript before publishing:

Line 40: the authors claimed that "soil microbial community structures are not significantly altered by mechanical weeding", this was claimed in the abstract and the discussion (lines 325, 341)? However, after reading the manuscript I didn’t find any statistical analysis that supports this claim “significantly altered”?? It was also mentioned throughout the manuscript (lines 206, 234, 252, 269, 328, 348, 350) that mechanical weeding altered the microbial diversity of the soil, yet the final conclusion claimed that these reported changes in the microbial diversity were not “significant”… authors need to support and explain this conclusion with statistical analysis.

The experiment included three replicate blocks (lines 108-112) and that was good for proper statistical analysis for the soil biological properties, yet in the soil microbial diversity analysis the replicates were pooled (line 152) and thus no statistical analysis was done to compare the two groups (no weeding and mechanical weeding) throughout the manuscript.

Line 163: The authors mentioned that they used “the free online Majorbio Cloud Platform”? The authors should mention in details which tools in the mentioned platform were used for the microbiome analysis… was it QIIME or Mothur or what??? They should also mention which database was used for taxonomy picking and alignment…. was it Greengages or SILVA or what?? they should also mention the version used in the analysis.

Figures 1, 2 and 3: the tool/software used to generate these figures should be mentioned in the legend.

Tables 3 and 4: These tables represent the same data in figure 2, they may move to supplementary materials to avoid redundancy in the manuscript.

Line 251: The results showed that two fungal genera were unique in the NW (no weeding ) soil, these two genera didn’t appear in the MW (mechanical weeding) soil, yet throughout the manuscript the authors didn’t rationalize this finding or discussed the functionality of these two genera (are they plant growth-promoting fungi or toxic fungi?? )…. this need to be addressed

For better contrasting the two soil groups (NW and MW) the authors might check the core microbial community in each group and see whether it is different or not… many free tools are available for this analysis (LEfSe, QIIME,..)

Reviewer #2: In the manuscript by Xiao et al., the authors measure several properties of soil fertility (the activities of three extracellular enzymes and microbial biomass C, N and P) and examine bacterial and fungal taxa in agricultural fields of star anise that were subjected to mechanical weeding. These same soil properties were measured in control plots that were not weeded and compared between the two treatments. The authors conclude that the soil microbial community structure is not altered by mechanical weeding; however, the lack of statistical analysis on the bacterial and fungal community data make such a conclusion impossible, in my opinion. Further, the authors conclude that mechanical weeding improve soil health, but I don’t think the phosphatase and microbial biomass P data support this conclusion. My comments are detailed below. But, overall, I suggest a more rigorous investigation of the data, especially the microbial community data.

Overall concerns:

The conclusions made about bacterial and fungal community changes are based on proportional representation of different phyla or genera. But, there are no statistics conducted on these numbers, despite the authors concluding that no significant differences were found at the phylum level (lines 40 and 326). The authors also make conclusions about genera that change between their treatments (line 347), again without statistics being done on the data. There are programs available that can run t-tests between treatments with corrections for multiple comparisons to test for differences between specific taxa (see STAMP for example https://doi.org/10.1007/978-1-4614-6418-1_780-1). But, what I suggest for this data set is that the authors explore the metagenomic data further before making conclusions, such as by calculating alpha diversity (richness, Shannon diversity, etc.) and beta diversity (i.e., turnover). These will help with whether or not you can conclude that the MW treatment led to “less changed” communities than the NW treatment (lines 349-350). Community data like what are presented here are also typically analyzed with multivariate statistics, such as ordination (principle coordinates analysis, nonmetric multidimensional scaling, etc.) and permutational multivariate analysis of variance (PERMANOVA). Fungal taxa can be assigned to different guilds using FUNGuild (Nguyen et al. 2016, doi: 10.1016/j.funeco.2015.06.006) or the newly available FungalTraits (Polme et al. 2020, doi: 10.1007/s13225-020-00466-2). Exploring how the different fungal guilds changed between treatments could be useful, especially because the Discussion mentions beneficial plant symbionts (i.e., mycorrhizal fungi) (lines 308-311).

The phosphatase activity and microbial biomass P data presented here suggest an increase in P demand in the MW plots. The increase in phosphatase activity and decrease in microbial biomass P with weeding, to me, suggests that the organisms were actively acquiring P from organic sources to a greater extent, potentially because inorganic P availability was lower with weeding. Overall, it was difficult for me to follow the conclusions the authors make based on the extracellular enzyme activity and microbial biomass C, N and P data because they don’t compare their numbers with other experiments. What have other studies found with mechanical weeding and how it affects extracellular enzyme activity and assimilated nutrients into microbial biomass? In particular, the difference in microbial biomass C between NW and MW plots reported here (Table 2) is quite large. Is this typical for mechanical weeding? Placing the numbers reported here into context by relating them to other published studies is needed.

Minor comments:

Line 67: Do weeds really compete with agricultural plants for oxygen? I skimmed reference 12 and don’t see anything about weeds competing with crops for oxygen.

Line 110: Was each plot divided in half - one side weeded one side not weeded? A methods figure could help here with a diagram of the set up.

Line 115: Each species of what? Aren't all plant species star anise? Please clarify in the methods.

Lines 162-164: It seems to me that Majorbio is just a cloud for processing data. What pipeline or process was used for the sequence reads? More details are needed on how chimeras were removed, how low-quality reads were filtered out, at what percent OTUs were clustered, etc. If chimeras and low-quality reads were not removed, the conclusions for microbial taxa could be based on sequence errors. The process for analyzing sequence reads needs to be explained in the methods.

Lines 300-301: I don't follow the conclusion that weeding is not completely harmful to soil fertility. The phosphatase activity and microbial biomass P data suggest increased P demand (i.e., lower P availability) with mechanical weeding. Wouldn’t this be a harmful effect of weeding? P is an essential nutrient to plants. More context is needed on how the enzyme and microbial biomass data compare to other agricultural studies.

Line 307: Fungi are decomposers in more ecosystems than just forests. Please rephrase.

6. PLOS authors have the option to publish the peer review history of their article (what does this mean?). If published, this will include your full peer review and any attached files.

Reviewer #1: **Yes: **Mariam Hassan

Reviewer #2: No

---

## [Author Response · Author response to Decision Letter 0]

4 Jan 2022

Dear Editors and Reviewers:

Thank you for your letter and the reviewers’ comments concerning our manuscript entitled “Effects of mechanical weeding on soil fertility and microbial community structure in star anise (Illicium verum Hook.f.) plantations” (PONE-D-21-29819). Those comments are all valuable and very helpful for revising and improving our paper, as well as the important guiding significance tour researches. We have read comments carefully and have made correction which we hope meet with approval. Revised portion are marked in Green in the paper. The main corrections in the paper and the responds to the reviewer’s comments are as flowing:

Responds to the Editors’ and Reviewers’ comments:

Journal Requirements:

Reply: The authors have revised the manuscript style according to the PLOS ONE style templates.

Reply: No permits were required. Because the experimental site belongs to the national forest farm, and it is one of the experimental sites of Guangxi Forestry Research Institute.

“This work was supported by Open Research Fund of Guangxi Key Laboratory of Special Non-wood Forest Cultivation & Utilization (19-B-02-01), Project of key R & D of Guangxi (No. AB1850014) and the eighth batch of special funds for specially invited experts in Guangxi Province, China.”

Reply: This work was mainly supported by Open Research Fund of Guangxi Key Laboratory of Special Non-wood Forest Cultivation & Utilization (19-B-02-01). However, the money in this fund was not wholly enough to carry out the whole experiment. Therefore, the authors used part of the money in project 2 and 3 for conducting the experiment continuously. The funders had no role in study design, data collection and analysis, decision to publish, or preparation of the manuscript.

Reply: The corresponding author have linked an ORCID iD to his Editorial Manager account.

Additional Editor Comments:

There appears to be an overall lack of rigor in the experimental setup and data analysis performed. In particular, the authors must include more detailed statistical analysis and additional analyses performed to support the conclusions made.

There appears to be an overall lack of rigor in the experimental setup and data analysis performed. In particular, the authors must include more detailed statistical analysis and additional analyses performed to support the conclusions made.

Reply: The authors have revised the content in manuscript according to the Editor comments.

Reviewers' comments:

Reviewer's Responses to Questions

Comments to the Author

1. Is the manuscript technically sound, and do the data support the conclusions?

Reviewer #1: Partly

Reviewer #2: Partly

Reply: The authors have revised this part in manuscript according to the Reviewers’ comments.

2. Has the statistical analysis been performed appropriately and rigorously?

Reviewer #1: No

Reviewer #2: No

Reply: The authors have revised this part in manuscript according to the Reviewers’ comments.

3. Have the authors made all data underlying the findings in their manuscript fully available?

Reviewer #1: Yes

Reviewer #2: No

4. Is the manuscript presented in an intelligible fashion and written in standard English?

Reviewer #1: Yes

Reviewer #2: Yes

5. Review Comments to the Author

Reviewer #1: 

The manuscript is well organized and written but there are some major concerns that need to be addressed in the manuscript before publishing:

Line 40: the authors claimed that "soil microbial community structures are not significantly altered by mechanical weeding", this was claimed in the abstract and the discussion (lines 325, 341)? However, after reading the manuscript I didn’t find any statistical analysis that supports this claim “significantly altered”?? It was also mentioned throughout the manuscript (lines 206, 234, 252, 269, 328, 348, 350) that mechanical weeding altered the microbial diversity of the soil, yet the final conclusion claimed that these reported changes in the microbial diversity were not “significant”… authors need to support and explain this conclusion with statistical analysis.

Reply: The authors have revised this part in manuscript according to the Reviewers’ comments.

The experiment included three replicate blocks (lines 108-112) and that was good for proper statistical analysis for the soil biological properties, yet in the soil microbial diversity analysis the replicates were pooled (line 152) and thus no statistical analysis was done to compare the two groups (no weeding and mechanical weeding) throughout the manuscript.

Reply: The authors have revised this part in manuscript according to the Reviewers’ comments.

Line 163: The authors mentioned that they used “the free online Majorbio Cloud Platform”? The authors should mention in details which tools in the mentioned platform were used for the microbiome analysis… was it QIIME or Mothur or what??? They should also mention which database was used for taxonomy picking and alignment…. was it Greengages or SILVA or what?? they should also mention the version used in the analysis.

Reply: The authors have revised this part in manuscript according to the Reviewers comments.

Figures 1, 2 and 3: the tool/software used to generate these figures should be mentioned in the legend.

Reply: The authors had revised the points where Reviewers suggested in the manuscript. And, the tool/software were also used to generate the figures mentioned in manuscript.

Tables 3 and 4: These tables represent the same data in figure 2, they may move to supplementary materials to avoid redundancy in the manuscript.

Reply: The authors had revised the points where Reviewers suggested in the manuscript.

Line 251: The results showed that two fungal genera were unique in the NW (no weeding ) soil, these two genera didn’t appear in the MW (mechanical weeding) soil, yet throughout the manuscript the authors didn’t rationalize this finding or discussed the functionality of these two genera (are they plant growth-promoting fungi or toxic fungi?? )…. this need to be addressed

Reply: The authors have revised this content in manuscript according to the Reviewers’ comments.

For better contrasting the two soil groups (NW and MW) the authors might check the core microbial community in each group and see whether it is different or not… many free tools are available for this analysis (LEfSe, QIIME.)

Reply: The authors have revised this content according to the Reviewers suggestions. Also, the authors analyzed the microbial communities of the two soil treatments by the LEfSe tool.

Special thanks to you for your good comments.

Reviewer #2:

In the manuscript by Xiao et al., the authors measure several properties of soil fertility (the activities of three extracellular enzymes and microbial biomass C, N and P) and examine bacterial and fungal taxa in agricultural fields of star anise that were subjected to mechanical weeding. These same soil properties were measured in control plots that were not weeded and compared between the two treatments. The authors conclude that the soil microbial community structure is not altered by mechanical weeding; however, the lack of statistical analysis on the bacterial and fungal community data make such a conclusion impossible, in my opinion. Further, the authors conclude that mechanical weeding improve soil health, but I don’t think the phosphatase and microbial biomass P data support this conclusion. My comments are detailed below. But, overall, I suggest a more rigorous investigation of the data, especially the microbial community data.

Reply: The authors have revised this content in manuscript according to the Reviewers’ comments.

Overall concerns:

The conclusions made about bacterial and fungal community changes are based on proportional representation of different phyla or genera. But there are no statistics conducted on these numbers, despite the authors concluding that no significant differences were found at the phylum level (lines 40 and 326). The authors also make conclusions about genera that change between their treatments (line 347), again without statistics being done on the data. There are programs available that can run t-tests between treatments with corrections for multiple comparisons to test for differences between specific taxa (see STAMP for example https://doi.org/10.1007/978-1-4614-6418-1_780-1). But, what I suggest for this data set is that the authors explore the metagenomic data further before making conclusions, such as by calculating alpha diversity (richness, Shannon diversity, etc.) and beta diversity (i.e., turnover). These will help with whether or not you can conclude that the MW treatment led to “less changed” communities than the NW treatment (lines 349-350). Community data like what are presented here are also typically analyzed with multivariate statistics, such as ordination (principle coordinates analysis, nonmetric multidimensional scaling, etc.) and permutational multivariate analysis of variance (PERMANOVA). Fungal taxa can be assigned to different guilds using FUNGuild (Nguyen et al. 2016, doi: 10.1016/j.funeco.2015.06.006) or the newly available FungalTraits (Polme et al. 2020, doi: 10.1007/s13225-020-00466-2). Exploring how the different fungal guilds changed between treatments could be useful, especially because the Discussion mentions beneficial plant symbionts (i.e., mycorrhizal fungi) (lines 308-311).

Reply: The authors have revised these points according to Reviewers’ suggestion in manuscript. That is, using the FUNGuild and the LEfSe tools in manuscript analyzed the functional predictions of the microbial community.

The phosphatase activity and microbial biomass P data presented here suggest an increase in P demand in the MW plots. The increase in phosphatase activity and decrease in microbial biomass P with weeding, to me, suggests that the organisms were actively acquiring P from organic sources to a greater extent, potentially because inorganic P availability was lower with weeding. Overall, it was difficult for me to follow the conclusions the authors make based on the extracellular enzyme activity and microbial biomass C, N and P data because they don’t compare their numbers with other experiments. What have other studies found with mechanical weeding and how it affects extracellular enzyme activity and assimilated nutrients into microbial biomass? In particular, the difference in microbial biomass C between NW and MW plots reported here (Table 2) is quite large. Is this typical for mechanical weeding? Placing the numbers reported here into context by relating them to other published studies is needed.

Reply: The authors have revised this content in manuscript according to the Reviewers’ comments.

Minor comments:

Line 67: Do weeds really compete with agricultural plants for oxygen? I skimmed reference 12 and don’t see anything about weeds competing with crops for oxygen.

Reply: The authors have revised this content in manuscript according to the Reviewers’ comments.

Line 110: Was each plot divided in half - one side weeded one side not weeded? A methods figure could help here with a diagram of the set up.

Reply: The authors have revised this content in manuscript according to the Reviewers’ comments.

Line 115: Each species of what? Aren't all plant species star anise? Please clarify in the methods.

Reply: The authors re-checked this part and revised part of the content in manuscript according to the Reviewers’ comments.

Lines 162-164: It seems to me that Majorbio is just a cloud for processing data. What pipeline or process was used for the sequence reads? More details are needed on how chimeras were removed, how low-quality reads were filtered out, at what percent OTUs were clustered, etc. If chimeras and low-quality reads were not removed, the conclusions for microbial taxa could be based on sequence errors. The process for analyzing sequence reads needs to be explained in the methods.

Reply: The authors re-checked this part and revised the content in manuscript according to the Reviewers’ comments.

Lines 300-301: I don't follow the conclusion that weeding is not completely harmful to soil fertility. The phosphatase activity and microbial biomass P data suggest increased P demand (i.e., lower P availability) with mechanical weeding. Wouldn’t this be a harmful effect of weeding? P is an essential nutrient to plants. More context is needed on how the enzyme and microbial biomass data compare to other agricultural studies.

Reply: The authors re-checked this part and revised part of the content in manuscript according to the Reviewers’ comments.

Line 307: Fungi are decomposers in more ecosystems than just forests. Please rephrase.

Reply: The authors re-checked this part and revised part of the content in manuscript according to the Reviewers’ comments.

Special thanks to you for your good comments.

6. PLOS authors have the option to publish the peer review history of their article (what does this mean?). If published, this will include your full peer review and any attached files.

Do you want your identity to be public for this peer review? For information about this choice, including consent withdrawal, please see our Privacy Policy.

Reviewer #1: Yes: Mariam Hassan

Reviewer #2: No

Special thanks to you for your good comments.

We tried our best to improve the manuscript and made some changes will not influence the content and framework of the paper. We appreciate for Editors/Reviewers’ warm work earnestly, and hope that the correction will meet with approval.

Once again, thank you very much for your comments and suggestions.

Correspondence should be addressed to Prof.& Dr. Shang-dong Yang at the following address, phone number, and email address.

Corresponding author: 1) Name: Wenhui Liang; E-mail address: 723746615@qq.com; Tel: 86-771-2319966; 2) Name: Shangdong Yang; E-mail address: 924433816@qq.com; Tel: 86-771-3270813.

Thanks very much for your attention and consideration. We are looking forward to hearing from you soon.

With kind personal regards,

Sincerely yours,

Jian Xiao (E-mail: 1318513279@qq.com)

---

## [Decision Letter · Decision Letter 1]

10 Mar 2022

PONE-D-21-29819R1

Effects of mechanical weeding on soil fertility and microbial community structure in star anise (Illicium verum Hook.f.) plantations

PLOS ONE

Dear Dr. Yang,

Thank you for submitting your manuscript to PLOS ONE. After careful consideration, we feel that it has merit but does not fully meet PLOS ONE’s publication criteria as it currently stands. Therefore, we invite you to submit a revised version of the manuscript that addresses the points raised during the review process. In particular, you all have not adequately responded to the reviewers' concerns regarding performing appropriate statistical analyses on the data, such that the conclusions are adequately supported statistically. I will remind the authors that it is important to solid statistical support to ensure scientific rigor. I will allow one more round of revision for the authors to address these issues raised by the reviewers.

We look forward to receiving your revised manuscript.

Kind regards,

Brenda A Wilson, Ph.D.

Academic Editor

PLOS ONE

Additional Editor Comments:

At this point, the authors have still not included appropriate statistics to support their conclusions despite several suggestions by the reviewers, and so the majority of their conclusions are not supported with appropriate statistics.

Reviewers' comments:

Reviewer's Responses to Questions

**Comments to the Author**

1. If the authors have adequately addressed your comments raised in a previous round of review and you feel that this manuscript is now acceptable for publication, you may indicate that here to bypass the “Comments to the Author” section, enter your conflict of interest statement in the “Confidential to Editor” section, and submit your "Accept" recommendation.

Reviewer #1: (No Response)

Reviewer #2: (No Response)

2. Is the manuscript technically sound, and do the data support the conclusions?

Reviewer #1: Yes

Reviewer #2: No

3. Has the statistical analysis been performed appropriately and rigorously? 

Reviewer #1: No

Reviewer #2: No

4. Have the authors made all data underlying the findings in their manuscript fully available?

Reviewer #1: Yes

Reviewer #2: Yes

5. Is the manuscript presented in an intelligible fashion and written in standard English?

Reviewer #1: Yes

Reviewer #2: Yes

6. Review Comments to the Author

Reviewer #1: Regarding the statistical analysis, throughout the manuscript the authors didn’t mention the used tests to analyse the data. As a reader and reviewer we need to know the used statistical test and wether it was appropriate or not. Wherever you mention any significance with P value you need to mention the used statistical test.

Reviewer #2: While I think the data collected by the authors for this manuscript are valuable, the study still lacks proper statistical analysis for the conclusions that are being made. I do see that both FUNGuild and LEfSe were used to analyze the data, but a large portion of the Results and Conclusions are still based on changes in the abundances of different taxa and no statistics are conducted on these numbers.

Lines 235-302: These results and the conclusions that follow are all just observations on how the abundance of different taxa change. Statistics need to be conducted to make conclusions about how the taxa changed. As I suggested previously, I recommend using the free software STAMP (https://doi.org/10.1007/978-1-4614-6418-1_780-1). The website is here: https://beikolab.cs.dal.ca/software/STAMP. This program will allow you to conduct t-tests between the MW and NW treatments. Both taxonomic identities and functional groups from FUNGuild can be compared (lines 341-348).

Lines 305-307: There are plenty of statistics that do analyze community data at the OTU level. As I suggested previously, community data like what are presented here are also typically analyzed with multivariate statistics, such as ordination (principle coordinates analysis, nonmetric multidimensional scaling, etc.) and permutational multivariate analysis of variance (PERMANOVA). These statistics can all be run in the vegan package of the program R. You can download R for free here: https://www.r-project.org/.

Discussion: I previously commented "The phosphatase activity and microbial biomass P data suggest increased

P demand (i.e., lower P availability) with mechanical weeding. Wouldn’t this be a harmful effect of weeding? P is an essential nutrient to plants. More context is needed on how the enzyme and microbial biomass data compare to other agricultural studies." In the revised manuscript, I don't see any inclusion of extracellular enzyme production or microbial biomass nutrient content from other agricultural studies. This will help put the data presented in the current study into context. in addition, I do not see any discussion of changes to P availability with mechanical weeding. Given the increase in phosphatase production and decrease in microbial biomass P with mechanical weeding, I think that there is lower P availability with mechanical weeding. The way I interpret the data presented, the mechanical weeding increases assimilation of nutrients by the soil microbes (thus the increase in organic C and organic N content). But, organic P content does not show this same pattern. Instead, phosphatase activity increases with mechanical weeding. This means that there is not enough inorganic P available, so the microbes need to put their energy into releasing phosphatase to free up P that is bound in organic matter. In other words, the data suggest that there is a higher P demand or lower P availability with mechanical weeding. I understand that the authors may disagree with me here, but the Discussion overall lacks these kinds of conclusions and mostly just re-states the results.

7. PLOS authors have the option to publish the peer review history of their article (what does this mean?). If published, this will include your full peer review and any attached files.

Reviewer #1: **Yes: **Mariam Hassan

Reviewer #2: No

---

## [Author Response · Author response to Decision Letter 1]

13 Mar 2022

Dear Editors and Reviewers:

Thank you for your letter and the reviewers’ comments concerning our manuscript entitled “Effects of mechanical weeding on soil fertility and microbial community structure in star anise (Illicium verum Hook.f.) plantations” (PONE-D-21-29819). Those comments are all valuable and very helpful for revising and improving our paper, as well as the important guiding significance tour researches. We have read comments carefully and have made correction which we hope meet with approval. Revised portion are marked in Green in the paper. The main corrections in the paper and the responds to the reviewer’s comments are as flowing:

Responds to the Editors’ and Reviewers’ comments:

Additional Editor Comments:

At this point, the authors have still not included appropriate statistics to support their conclusions despite several suggestions by the reviewers, and so the majority of their conclusions are not supported with appropriate statistics.

Reply: The authors have revised the content in manuscript according to the Editor comments.

Reviewers' comments:

Reviewer #1: 

Regarding the statistical analysis, throughout the manuscript the authors didn’t mention the used tests to analyse the data. As a reader and reviewer we need to know the used statistical test and wether it was appropriate or not. Wherever you mention any significance with P value you need to mention the used statistical test.

Reply: The authors have revised the content in manuscript according to the Reviewer comments.

Special thanks to you for your good comments.

Reviewer #2: 

While I think the data collected by the authors for this manuscript are valuable, the study still lacks proper statistical analysis for the conclusions that are being made. I do see that both FUNGuild and LEfSe were used to analyze the data, but a large portion of the Results and Conclusions are still based on changes in the abundances of different taxa and no statistics are conducted on these numbers.

Reply: LEfSe is a software for discovering high-dimensional biomarkers and revealing genomic features to distinguish between two or more biological conditions (or taxa). The software first uses the non-parametric factorial Kruskal-Wallis (KW) sum-rank test (non-parametric factorial Kruskal-Wallis rank sum test) to detect features with significant abundance differences and to find taxa that differ significantly from the abundance. Finally, LEfSe used linear discriminant analysis (LDA) to estimate the magnitude of the effect of each component (species) abundance on the difference effect. Therefore, we can directly identify bacteria and fungi with significant differences in NW and MW by LEfSe directly, without the need to do a separate test for differences between groups.

In addition, we had used R language (version 3.3.1) to perform Wilcoxon rank-sum test and graphs, compared taxonomic identities and functional groups from FUNGuild of NW and MW.

Lines 235-302: These results and the conclusions that follow are all just observations on how the abundance of different taxa change. Statistics need to be conducted to make conclusions about how the taxa changed. As I suggested previously, I recommend using the free software STAMP (https://doi.org/10.1007/978-1-4614-6418-1_780-1). The website is here: https://beikolab.cs.dal.ca/software/STAMP. This program will allow you to conduct t-tests between the MW and NW treatments. Both taxonomic identities and functional groups from FUNGuild can be compared (lines 341-348).

Reply: We have used R language (version 3.3.1) to perform Wilcoxon rank-sum test and graphs, compared taxonomic identities and functional groups from FUNGuild of NW and MW.

Lines 305-307: There are plenty of statistics that do analyze community data at the OTU level. As I suggested previously, community data like what are presented here are also typically analyzed with multivariate statistics, such as ordination (principle coordinates analysis, nonmetric multidimensional scaling, etc.) and permutational multivariate analysis of variance (PERMANOVA). These statistics can all be run in the vegan package of the program R. You can download R for free here: https://www.r-project.org/.

Reply: We have added Principal Component Analysis (PCA) to analyze community data at the OTU level. And R language (version 3.3.1) for PCA statistical analysis and graphing also was used. 

Discussion: I previously commented "The phosphatase activity and microbial biomass P data suggest increased P demand (i.e., lower P availability) with mechanical weeding. Wouldn’t this be a harmful effect of weeding? P is an essential nutrient to plants. More context is needed on how the enzyme and microbial biomass data compare to other agricultural studies." In the revised manuscript, I don't see any inclusion of extracellular enzyme production or microbial biomass nutrient content from other agricultural studies. This will help put the data presented in the current study into context. in addition, I do not see any discussion of changes to P availability with mechanical weeding. Given the increase in phosphatase production and decrease in microbial biomass P with mechanical weeding, I think that there is lower P availability with mechanical weeding. The way I interpret the data presented, the mechanical weeding increases assimilation of nutrients by the soil microbes (thus the increase in organic C and organic N content). But, organic P content does not show this same pattern. Instead, phosphatase activity increases with mechanical weeding. This means that there is not enough inorganic P available, so the microbes need to put their energy into releasing phosphatase to free up P that is bound in organic matter. In other words, the data suggest that there is a higher P demand or lower P availability with mechanical weeding. I understand that the authors may disagree with me here, but the Discussion overall lacks these kinds of conclusions and mostly just re-states the results.

Reply: Some types of weeds have good phosphorus uptake capacity, which can accumulate soil surplus phosphorus in biomass and promote phosphorus biocycling (Roy 2017). However, without total topsoil inversion, the plant materials cut by a brush cutter and incorporation into the soil which provide fertility is limited. A stratified layer of phosphorus builds up on or near the soil surface, increasing the risk of phosphorus release and subsequent transport of dissolved reactive phosphorus via surface runoff and tile drain water (Ulén et al, 2010). We speculate that this may be the reason why microbial biomass phosphorus was significantly lower in the MW treatment than in the NW treatment. 

Roy ED. Phosphorus recovery and recycling with ecological engineering: A review. Ecol Eng. 2017;98: 213-227.

Ulén B, Aronsson H, Bechmann M, Krogstad T, ØYgarden L, Stenberg M. Soil tillage methods to control phosphorus loss and potential side-effects: a Scandinavian review. Soil Use Manage. 2010; 26(2): 94-107.

Special thanks to you for your good comments.

We tried our best to improve the manuscript and made some changes will not influence the content and framework of the paper. We appreciate for Editors/Reviewers’ warm work earnestly, and hope that the correction will meet with approval.

Once again, thank you very much for your comments and suggestions.

Correspondence should be addressed to Prof.& Dr. Shang-dong Yang at the following address, phone number, and email address.

Corresponding author: 1) Name: Wenhui Liang; E-mail address: 723746615@qq.com; Tel: 86-771-2319966; 2) Name: Shangdong Yang; E-mail address: 924433816@qq.com; Tel: 86-771-3270813.

Thanks very much for your attention and consideration. We are looking forward to hearing from you soon.

With kind personal regards,

Sincerely yours,

Jian Xiao (E-mail: 1318513279@qq.com)

---

## [Decision Letter · Decision Letter 2]

31 Mar 2022

Effects of mechanical weeding on soil fertility and microbial community structure in star anise (Illicium verum Hook.f.) plantations

PONE-D-21-29819R2

Dear Dr. Yang,

We’re pleased to inform you that your manuscript has been judged scientifically suitable for publication and will be formally accepted for publication once it meets all outstanding technical requirements.

Kind regards,

Brenda A Wilson, Ph.D.

Academic Editor

PLOS ONE

Reviewers' comments:

Reviewer's Responses to Questions

**Comments to the Author**

1. If the authors have adequately addressed your comments raised in a previous round of review and you feel that this manuscript is now acceptable for publication, you may indicate that here to bypass the “Comments to the Author” section, enter your conflict of interest statement in the “Confidential to Editor” section, and submit your "Accept" recommendation.

Reviewer #1: All comments have been addressed

Reviewer #3: (No Response)

2. Is the manuscript technically sound, and do the data support the conclusions?

Reviewer #1: Yes

Reviewer #3: Yes

3. Has the statistical analysis been performed appropriately and rigorously? 

Reviewer #1: Yes

Reviewer #3: Yes

4. Have the authors made all data underlying the findings in their manuscript fully available?

Reviewer #1: Yes

Reviewer #3: Yes

5. Is the manuscript presented in an intelligible fashion and written in standard English?

Reviewer #1: Yes

Reviewer #3: Yes

6. Review Comments to the Author

Reviewer #1: (No Response)

Reviewer #3: I think the current version of the paper includes enough statistics (principal component analysis, linear discriminant analysis, and Wilcoxon rank-sum test) to support the conclusions of the authors. As a minor comment, I have only the following typo:

Line 185: substitute “tool, The” with “tool. The”.

7. PLOS authors have the option to publish the peer review history of their article (what does this mean?). If published, this will include your full peer review and any attached files.

Reviewer #1: **Yes: **Mariam Hassan

Reviewer #3: No

---

## [Editor Report · Acceptance letter]

4 Apr 2022

PONE-D-21-29819R2 

Effects of mechanical weeding on soil fertility and microbial community structure in star anise (*Illicium verum* Hook.f.) plantations 

Dear Dr. Yang:

I'm pleased to inform you that your manuscript has been deemed suitable for publication in PLOS ONE. Congratulations! Your manuscript is now with our production department. 

Kind regards, 

on behalf of

Dr. Brenda A Wilson 

Academic Editor

PLOS ONE